# Towards Neural No-Resource Language Translation: A Comparative Evaluation of Approaches

## Abstract

No-resource languages—those with minimal or no digital representation—pose unique challenges for machine translation (MT). Unlike low-resource languages, which rely on limited but existent corpora, no-resource languages often have fewer than 100 sentences available for training. This work explores the problem of no-resource translation through three distinct workflows: fine-tuning of translation-specific models, in-context learning with large language models (LLMs) using chain-of-reasoning prompting, and direct prompting without reasoning. Using Owens Valley Paiute as a case study, we demonstrate that no-resource translation demands fundamentally different approaches from low-resource scenarios, as traditional approaches to machine translation, such as those that work for low-resource languages, fail. Empirical results reveal that, although traditional approaches fail, the in-context learning capabilities of general-purpose large language models enable no-resource language translation that outperforms low-resource translation approaches and rivals human translations (BLEU 0.45-0.6); specifically, chain-of-reasoning prompting outperforms other methods for larger corpora, while direct prompting exhibits advantages in smaller datasets. As these approaches are language-agnostic, they have potential to be generalized to translation tasks from a wide variety of no-resource languages without expert input. These findings establish no-resource translation as a distinct paradigm requiring innovative solutions, providing practical and theoretical insights for language preservation.

## 1 Introduction

### 1.1 Background and Motivation

The advancement of machine translation (MT) has been marked by significant advances in neural architectures such as Transformers Vaswani et al. (2017). High-resource languages have benefited immensely from these breakthroughs because of the availability of large corpora. However, low-resource translation continues to pose significant challenges, addressed primarily through techniques such as data augmentation and transfer learning Sennrich et al. (2016); Fadaee et al. (2017). These methods rely on the presence of some foundational datasets, albeit small, to enhance translation performance.

A growing area of interest lies in no-resource languages, which we formally define as languages with minimal digital representation, specifically fewer than 100 documented phrases or sentences in any corpus Coleman et al. (2024). This category of languages represents a unique challenge: the absence of sufficient data renders traditional techniques like transfer learning and back-translation Zoph et al. (2016); Sennrich et al. (2016) ineffective. We hypothesized that traditional methods, such as fine-tuning large models Chowdhery et al. (2022), would fail due to the small corpus size, while the emergent reasoning capabilities of large language models (LLMs) Brown et al. (2020); Radford et al. (2019) could offer an alternative pathway for no-resource language translation.

The motivation behind this study stems from an intuitive observation: a human translator can analyze a small corpus, identify patterns, and fill in the gaps to produce coherent translations. For

instance, given the translations of phrases "the cat sleeps" and "the dog runs," a human might infer the translation of "the dog sleeps" or "the cat flies." Similarly, we propose that the general reasoning capabilities of LLMs can enable these models to extrapolate patterns and generate translations in the absence of sufficient data.

Our aim is twofold: to rigorously evaluate whether neural methods, such as fine-tuning pre-trained translation models, chain-of-reasoning prompting, and direct prompting, can address the no-resource translation problem; and to analyze the behavior of these methods to establish a better theoretical and practical understanding of neural no-resource translation. To limit the scope of the paper, we focus only on one-way translation from no-resource language to English. We believe this is the more important direction for most practical applications, as it allows for all resources in the no-resource language to be made accessible to the wider world. In doing so, we formalize the no-resource translation paradigm and provide insights into its potential to preserve linguistic diversity. This paper contributes:

- A formal definition and exploration of the no-resource language translation problem, distinct from low-resource translation challenges.
- A rigorous evaluation of three neural methods: fine-tuning, chain-of-reasoning prompting, and direct prompting.
- A functional, generalizable, and high-quality no-resource language translation workflow

By defining the problem space and investigating the viability of neural approaches, this study lays a foundation for future work in addressing the translation of no-resource languages, an essential step toward preserving global linguistic diversity.

## 2 RELATED WORK

The field of machine translation (MT) has undergone significant transformations, evolving from rule-based systems to statistical methods and, more recently, to neural approaches Vaswani et al. (2017). The introduction of Transformer architectures marked a paradigm shift, enabling state-of-the-art performance across high-resource languages Devlin et al. (2019); Raffel et al. (2020). However, these advancements have primarily benefited languages with large corpora, leaving low-resource and no-resource languages underexplored Tiedemann & Scherrer (2020); Zhang et al. (2020).

### 2.1 NEURAL APPROACHES TO MACHINE TRANSLATION

Neural MT methods have dominated recent research due to their scalability and ability to capture nuanced linguistic patterns Sennrich et al. (2016); Nguyen et al. (2021b). While these methods excel in high-resource settings, their success hinges on the availability of substantial training data Bahdanau et al. (2015). Techniques such as back-translation Li et al. (2024); Edunov et al. (2018), data augmentation Fadaee et al. (2017); Pascual et al. (2021), and transfer learning Zoph et al. (2016); Nguyen et al. (2021a) have demonstrated efficacy in low-resource contexts, such as Icelandic and Sinhala Wu et al. (2016). These languages are often represented in the pretraining corpora of general-purpose LLMs, which exhibit zero-shot translation capabilities for them Brown et al. (2020); Radford et al. (2019).

### 2.2 NO-RESOURCE LANGUAGE TRANSLATION

In contrast, no-resource languages lack even minimal digital representation, making traditional MT techniques ineffective. Coleman et al. Coleman et al. (2024) define no-resource languages as those with fewer than 100 documented phrases, necessitating alternative approaches. Recent work has introduced rule-based frameworks augmented by LLMs for specific no-resource languages, such as Owens Valley Paiute. Coleman et al. explore the potential of structured translations over raw-text outputs, establishing an initial foundation for addressing these challenges. Other efforts have investigated synthetic data generation, though these are often limited to relatively larger low-resource languages Liu et al. (2020); Tan (2021); Sheng et al. (2021).

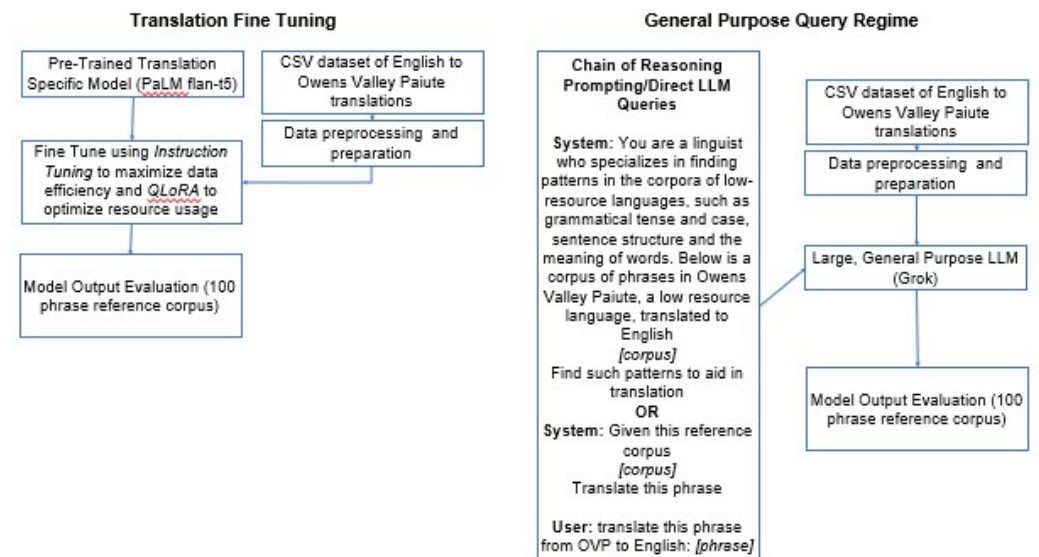

Figure 1: Comparative analysis of approaches for low-resource language translation: (left) a fine-tuning based methodology utilizing PaLM and QLORA, and (right) a general-purpose query regime leveraging large language models, which is used for both Chain-of-Reasoning and direct prompting experiments. Both approaches incorporate evaluation against a reference corpus for quality assurance.

### 2.3 NEURAL METHODS FOR NO-RESOURCE TRANSLATION

This study diverges from rule-based paradigms by investigating purely neural approaches for no-resource translation. Unlike previous work Coleman et al. (2024), which combines rule-based methodologies with LLM-assisted reasoning, our focus is on exploring fine-tuning, chain-of-reasoning prompting, and direct prompting as standalone strategies. Recent advancements in chain-of-reasoning prompting highlight its ability to enhance inferential tasks across domains Wei et al. (2022); Xie et al. (2022); Kojima et al. (2022). Similarly, direct prompting has shown promise in generating contextually accurate translations in constrained scenarios Ouyang et al. (2022; 2023). These methods leverage the emergent reasoning capabilities of LLMs, enabling inferences without requiring extensive linguistic rules.

The insights gained contribute not only to the theoretical understanding of LLMs in linguistically diverse scenarios but also to practical applications in language preservation.

## 3 METHODOLOGY

### 3.1 EXPERIMENTAL SETUP

Experiments were conducted using Owens Valley Paiute, a representative no-resource language with fewer than 100 documented phrases. Evaluation metrics included BLEU, ROUGE, TER, and METEOR, chosen for their ability to capture translation accuracy and semantic fidelity. The experimental setup involved testing different models and approaches using a standardized corpus for consistent comparison, as shown in Figure 1.

### 3.2 FINE-TUNING WITH QLORA

To evaluate the efficacy of methods proven to work for low-resource translation, we fine tuned PaLM, a translation LLM designed for fine tuning on low-resource corpora. Specifically, we applied QLoRA, a memory-efficient method for adapting large language models. The fine-tuning configuration utilized a low-rank adaptation with `r=8`, `lora_alpha=16`, and a dropout rate of 0.1. This

was targeted at the attention modules q and v, known for their role in contextual understanding. The model was then fine-tuned on the limited Owens Valley Paiute corpus using instruction tuning techniques, as is standard procedure for PaLM fine tuning Chowdhery et al. (2022). Finally, outputs were generated for phrases both in and not in the corpus.

### 3.3 CHAIN-OF-REASONING BASED PROMPTING

For in-context learning, we implemented a system prompt that presented the model with a small corpus of phrase translations, requesting it to infer the translation of novel phrases. The prompts leveraged chain-of-reasoning techniques to guide the model in deducing grammatical structures and semantic relationships. Prompts were submitted programmatically to the xAI API, with each translation request handled in an individual and isolated query to control for intra-session learning.

To test the model's reasoning capabilities, subsets of 10, 50, and 100 phrases were selected uniformly at random from the full 100-phrase corpus. For each phrase in a subset, we designed a unique prompt by first removing the target phrase from the corpus. The remaining phrases were included in the system prompt, framing them as known translations. The target phrase was then presented in the user prompt as the query for translation. This method emulated a real-world scenario where the model must infer a translation based on limited contextual information while ensuring the exclusion of direct hints from the input corpus.

This experimental setup ensured that the model was tested on its ability to generalize and reason, rather than simply memorizing patterns from the provided corpus. Metrics such as BLEU, ROUGE (1, 2, L), METEOR, and a normalized Translation Error Rate (TER) were used to evaluate translation quality. Results were visualized to highlight trends across corpus sizes, demonstrating the effectiveness of chain-of-reasoning prompts in addressing no-resource translation challenges.

### 3.4 DIRECT PROMPTING WITHOUT REASONING

The direct prompting method provided baseline results by using simple queries without explicit reasoning steps. As in the Chain-of-Reasoning based prompting approach, queries were made programmatically to the xAI API, and different corpus sizes were tested.

## 4 RESULTS

This section presents the experimental results for no-resource language translation using the three neural methods: fine-tuning, chain-of-reasoning prompting, and direct prompting. A list of translations produced by the prompting approaches can be found in Appendix A.

### 4.1 CHAIN-OF-REASONING PROMPTING

Chain-of-reasoning prompting demonstrated a significant aptitude for no-resource language translation, particularly for larger corpus sizes. Indeed, when provided the 99-word reference corpus, it had an average BLEU value of 0.48 As shown in Figure 2, performance metrics such as BLEU, ROUGE, and METEOR improved uniformly and consistently with increasing corpus sizes, plateauing near the 100-phrase mark.

### 4.2 DIRECT PROMPTING

Direct prompting exhibited human-level performance for small corpus sizes (BLEU 0.60) but struggled to generalize grammatical structures and infer vocabulary effectively as corpus sizes increased. For smaller corpus sizes (e.g., 10 phrases), direct prompting resulted in higher BLEU scores due to simple copying of the format and vocabulary of seen data, but performance decreased quickly (BLEU 0.47 on 99 word reference corpus), indicating limited inference capabilities when given much more data.

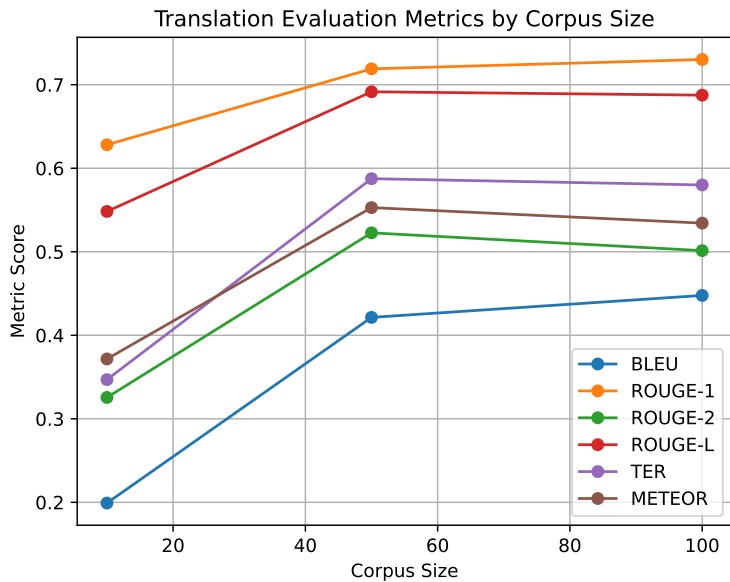

Figure 2: Performance scaling of chain-of-reasoning prompting across different corpus sizes. Metrics include BLEU, ROUGE, METEOR, and normalized TER.

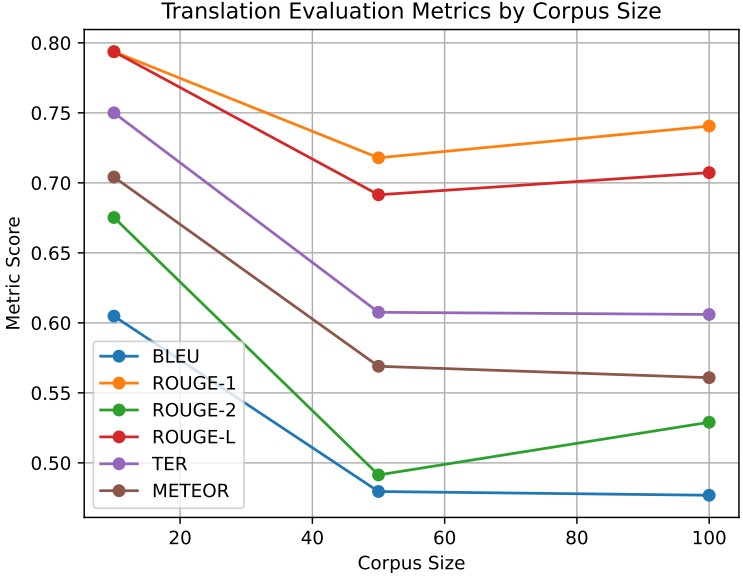

Figure 3: Performance scaling of direct prompting across different corpus sizes. Metrics include BLEU, ROUGE, METEOR, and normalized TER.

### 4.3 FINE-TUNING PRE-TRAINED MODELS

Fine-tuning smaller translation-specific models, such as PaLM Flan-T5 small V2, on OVP data yielded suboptimal results. BLEU scores for all corpus sizes were close to 0, reflecting a failure to produce meaningful translations. ROUGE-1 and ROUGE-2 scores also remained near 0, with TER scores exceeding 100, indicating high error rates. Notably, these models frequently outputted un-

translated text or translations in unrelated languages such as German or Turkish, failing to generalize to the target no-resource language.

# 5 DISCUSSION

The findings from this study underscore the unique challenges posed by no-resource languages, which are fundamentally distinct from low-resource translation tasks. Empirically, the failure of fine-tuning methods proven to work for low resource languages cements that the no-resource translation problem is fundamentally different as it lacks the critical mass of data required for low- and high- resource language translation. Chain-of-reasoning prompting stands out as the most promising solution, as it leverages emergent capabilities in LLMs to infer grammatical structures and semantic patterns effectively, while displaying notable capabilities generalizing to larger datasets. Importantly, all of these approaches were agnostic to the particular grammar and morphology of the no-resource language, suggesting that this could serve as a highly generalizable framework for translation from many low-resource languages.

## 5.1 THEORETICAL ALIGNMENT AND RAMIFICATIONS

These empirical results further highlight the theoretical distinction between low-resource and no-resource translation. Low-resource methods often rely on data augmentation Fadaee et al. (2017) or transfer learning Zoph et al. (2016), which presume some baseline corpus. By contrast, the absence of usable corpora for no-resource languages renders these approaches ineffective, necessitating entirely new methodologies. Compared to state-of-the-art approaches in low-resource contexts, which achieve BLEU scores exceeding 20% Sennrich et al. (2016), the performance achieved by chain-of-reasoning prompting (BLEU: 49%) represents a significant breakthrough for translation under extreme data scarcity.

From a theoretical perspective, these findings align with recent literature on emergent reasoning capabilities in LLMs Wei et al. (2022). The success of both the direct and chain-of-reasoning prompting experiments were enabled by a the in-context learning abilities of state-of-the-art LLMs Wei et al. (2022) The success of chain-of-reasoning prompting reinforces the hypothesis that inferential tasks benefit from large-scale pre-training, even in the absence of explicit linguistic representations for the target language. This suggests a paradigm shift in MT research, where general-purpose reasoning may supplant domain-specific fine-tuning for extremely low-resource scenarios.

## 5.2 TRANSLATOR OUTPUT ANALYSIS

The difference in corpus scaling between the direct and chain-of reasoning-prompting approaches demonstrate the fundamental difference in how translations are being generated between the two. Whereas the direct prompting approach proved to work better on small corpuses, where inference was not necessary and a more straightforward substitution and repetition approach worked, when the model was inundated with information, such as in the in the 50 - or 99-word corpuses, the approach showed worsening results, as proper translations require inference. On the other hand, the chain-of reasoning prompting demonstrated a consistently increasing performance across all metrics as corpus size increased. Specifically, this could be because of vocabulary acquisition - it was not guaranteed that the subsetted corpora provide the vocabulary necessary for the requested translation, or have said vocabulary in any significant volumes, and so the plateauing improvements likely show increased vocabulary acquisition. Indeed, patterns in data highlighted in Appendix A support this, as the bulk of the errors are in vocabulary, especially where there is ambiguity in the originals (e.g. original: "the bear cooked the wood", inferred translation: "the bear cooked this wood"). Even when vocabulary is wrong, similar words are often used, demonstrating high capabilities for understanding (e.g. original: "these are running", inferred translation: "these are playing"). Indeed, chain-of-reasoning prompting seems to serve as a robust translation method, where meaning is preserved even on the smallest reference corpora with scarce data, and improvements are shown with increasing corpus size.

Analysis of translation errors for each approach reveals consistent patterns, demonstrating the differential aptitudes of each translation approach. Fine-tuning methods struggled with basic vocabulary acquisition, often outputting the source language or unrelated text. Direct prompting showed

competency in regurgitating known phrases and formats but failed to generalize for novel inputs, particularly when grammatical inference was required. Chain-of-reasoning prompting, while the best-performing method, occasionally struggled with syntactic ambiguity, highlighting an area for future refinement.

### 5.3 FUTURE DIRECTIONS

Future work should explore several directions. First, synthetic data generation using advanced LLMs could expand the corpus size and improve performance across all methods. Indeed, it is possible that prompt-based no-resource translation approaches can be modified for effective corpus enlargement, allowing for more conventions MT approaches, such as those used for fine tuning. Further, extending the study to multiple no-resource languages from different linguistic families would validate the generalizability of these findings. Third, optimizing the design of system prompts for chain-of-reasoning could mitigate errors and further enhance scalability.

Finally, this study emphasizes the importance of interdisciplinary collaboration. Insights from computational linguistics, anthropology, and community engagement will be crucial in developing tools that not only perform well technically but also respect the cultural and linguistic nuances of endangered languages. The promising results achieved here mark a foundational step in addressing the challenges of no-resource language translation, with significant implications for both MT research and global linguistic preservation efforts.

## 6 CONCLUSION

No-resource languages present unique challenges that demand innovative solutions. This study demonstrates that while traditional MT methods fail, emergent capabilities in LLMs offer promising pathways for addressing these gaps. Specifically, this study has demonstrated that the in-context-learning capabilities of general purpose large language models can serve as an effective translation mechanism for no-resource languages. By defining the problem and exploring diverse workflows, this work provides a foundation for future research and practical implementations in preserving linguistic diversity.

## 7 DATA AND CODE ACCESSION

The dataset used in the experiment can be found at
`https://github.com/kubishi/yaduha/blob/main/data/translations.csv`.
The data, models and code used to fine tune PaLM and query the xAI API can be found at
`https://www.kaggle.com/code/standbackibyte/no-resource-tranlsation-fine-tuning-palm`
and
`https://www.kaggle.com/code/standbackibyte/no-resource-tranlsation-llm-queries`,
respectively.

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

## A  TRANSLATION OUTPUTS

This appendix provides detailed tables of the outputs from chain-of-reasoning and direct prompting experiments for corpus sizes of 10, 50, and 100 phrases.

### A.1  CHAIN-OF-REASONING PROMPTING OUTPUTS

Table 1: Translation outputs for chain-of-reasoning prompting experiments (10 phrases)

| Original Phrase | Translation |
|---|---|
| That lizard is going to write. | The lizard is going to write. |
| The bird snake climbed. | The snake climbed. |
| This finds this water. | This water finds these. |
| The bear cooked the wood. | The bear cooked this wood. |
| Those pinenuts climb. | The pinenuts climb. |
| They are going to write to us, you and I. | They will write us. |
| They are seeing these earths. | The earth is seeing/hearing those corn. |
| He/she/it reads the squirrel. | The squirrel is reading this. |
| That jackrabbit smiles. | The jackrabbit smiles. |
| They are hitting those chairs. | That chair is hitting them. |
| Metrics: BLEU: 0.199, ROUGE-1: 0.628, ROUGE-2: 0.326, ROUGE-L: 0.548 TER: 0.347, METEOR: 0.372 | |

Table 2: Translation outputs for chain-of-reasoning prompting experiments (50 phrases)

| Original Phrase | Translation |
|---|---|
| They visit us. | They are visiting us. |
| You and I will climb. | You and I will climb. |
| Water is going to talk to those chairs. | That water is going to talk to that chair. |
| This bear hears those worms. | The bear has heard those worms. |
| The wickiup is standing. | That wickiup is flying. |
| *Additional phrases omitted for brevity* | *Additional translations omitted for brevity* |
| Metrics: BLEU: 0.421, ROUGE-1: 0.719, ROUGE-2: 0.523, ROUGE-L: 0.692 TER: 0.588, METEOR: 0.553 | |

Table 3: Translation outputs for chain-of-reasoning prompting experiments (100 phrases)

| Original Phrase | Translation |
|---|---|
| This chair will see this rock. | This chair will see this rock. |
| These are running. | These are playing. |
| That lizard will fly. | The lizard will dance. |
| This dog sleeps. | The dog is sleeping. |
| We are sleeping. | We are sleeping. |
| *Additional phrases omitted for brevity* | *Additional translations omitted for brevity* |
| Metrics: BLEU: 0.448, ROUGE-1: 0.730, ROUGE-2: 0.501, ROUGE-L: 0.688 TER: 0.580, METEOR: 0.534 | |

### A.2  DIRECT PROMPTING OUTPUTS

Table 4: Translation outputs for direct prompting experiments (10 phrases)

| Original Phrase | Translation |
|---|---|
| The river is going to cook the mountain. | That river is going to cook this mountain. |
| The bird has heard those horses. | The bird has heard those horses. |
| That lizard has climbed. | The lizard has climbed. |
| This coyote is going to write these rocks. | This coyote is going to write to this rock. |
| Coffee is walking. | That coffee is dancing. |
| The tree has drunk the mountain. | That tree has drunk this mountain. |
| He/she/it is smiling. | He/she/it is smiling. |
| That lizard is going to write. | That lizard is going to write. |
| This tail is going to hit those wicki-ups. | This tail is going to hit this wickiup. |
| That lizard will fly. | The lizard will cry. |
| Metrics: BLEU: 0.605, ROUGE-1: 0.794, ROUGE-2: 0.675, ROUGE-L: 0.794 TER: 0.750, METEOR: 0.704 | |

Table 5: Translation outputs for direct prompting experiments (50 phrases)

| Original Phrase | Translation |
|---|---|
| That lizard will fly. | The lizard will spit. |
| This chair will visit those coffees. | The chair will visit those coffees. |
| This water is eating that food. | This water is eating those foods. |
| He/she/it is going to drink this water. | The water is going to drink the food. |
| This apple has eaten this wood. | The apple has eaten this wood. |
| *Additional phrases omitted for brevity* | *Additional translations omitted for brevity* |
| Metrics: BLEU: 0.480, ROUGE-1: 0.718, ROUGE-2: 0.491, ROUGE-L: 0.691 TER: 0.608, METEOR: 0.569 | |

Table 6: Translation outputs for direct prompting experiments (100 phrases)

| Original Phrase | Translation |
|---|---|
| The squirrel is hearing these apples. | The squirrel is hearing this apple. |
| You and I wrote. | They wrote. |
| This fish is eating those worms. | This fish is eating those worms. |
| The wickiup is standing. | The wickiup is flying. |
| This will hear these fish. | This fish is going to hear these. |
| *Additional phrases omitted for brevity* | *Additional translations omitted for brevity* |
| Metrics: BLEU: 0.477, ROUGE-1: 0.741, ROUGE-2: 0.529, ROUGE-L: 0.707 TER: 0.606, METEOR: 0.561 | |

