# OpenReview forum: "Towards Neural No-Resource Language Translation: A Comparative Evaluation of Approaches"
_ICLR.cc/2025/Workshop/BuildingTrust — Submitted to BuildingTrust_

### Official Review · Reviewer_5U7c · 2025-02-18
**This paper addresses the underexplored challenge of translating no-resource languages (fewer than 100 documented phrases) using neural methods. It rigorously evaluates three approaches: fine-tuning, chain-of-reasoning prompting, and direct prompting, with Owens Valley Paiute as a case study. The results demonstrate that traditional fine-tuning fails, while chain-of-reasoning prompting with large language models (LLMs) achieves human-level translation quality (BLEU 0.45-0.6), outperforming direct prompting for larger corpora. The work is original, well-structured, and significant for language preservation, though it could improve by expanding language coverage, exploring synthetic data generation, and addressing ethical concerns. Overall, it provides a strong foundation for future research in no-resource translation.**

**Rating:** 7
**Confidence:** 4

**Review:**

The paper is well-structured and methodologically sound. It addresses a significant gap in machine translation (MT) research by focusing on no-resource languages, which are often overlooked in favor of low-resource languages. The authors provide a clear definition of no-resource languages (fewer than 100 documented phrases) and rigorously evaluate three distinct approaches: fine-tuning, chain-of-reasoning prompting, and direct prompting. The empirical results are well-documented, with comprehensive metrics (BLEU, ROUGE, TER, METEOR) used to evaluate translation quality. The use of Owens Valley Paiute as a case study is appropriate, given its status as a no-resource language.

However, the paper could benefit from a more detailed discussion of the limitations of the study. For instance, the evaluation is limited to one language, and the generalizability of the findings to other no-resource languages is not thoroughly explored. Additionally, while the authors mention the potential for synthetic data generation, they do not provide concrete experiments or results in this direction.

The paper is generally clear and well-written. The introduction provides a strong motivation for the study, and the methodology is described in sufficient detail to allow for replication. The use of figures (e.g., Figures 1, 2, and 3) effectively illustrates the comparative performance of the different approaches. The discussion section is particularly strong, offering insights into the theoretical implications of the findings and suggesting future directions.

One area for improvement is the clarity of the experimental setup. While the authors describe the fine-tuning process and prompting methods, more details on the specific prompts used for chain-of-reasoning and direct prompting would be helpful. Additionally, the paper could benefit from a clearer explanation of why fine-tuning fails in no-resource scenarios, as opposed to low-resource ones.

The paper is highly original in its focus on no-resource languages, a niche but important area in MT research. The authors make a compelling case that no-resource translation is fundamentally different from low-resource translation and requires novel approaches. The use of chain-of-reasoning prompting is particularly innovative, leveraging the emergent reasoning capabilities of large language models (LLMs) to infer translations from minimal data.

The originality of the work is further highlighted by its departure from traditional rule-based or data-augmentation approaches, which are ineffective for no-resource languages. Instead, the authors explore purely neural methods, demonstrating that LLMs can achieve human-level translation quality without extensive linguistic rules or large corpora.

The study also contributes to the broader field of MT by challenging the assumption that large corpora are necessary for effective translation. The success of chain-of-reasoning prompting suggests that general-purpose reasoning capabilities in LLMs may supplant domain-specific fine-tuning in extreme low-resource scenarios.

Pros
Novel Focus: The paper addresses a critical gap in MT research by focusing on no-resource languages, which are often neglected in favor of low-resource languages.

Innovative Methods: The use of chain-of-reasoning prompting is a novel approach that leverages the emergent reasoning capabilities of LLMs to infer translations from minimal data.

Strong Empirical Results: The paper provides robust empirical evidence that chain-of-reasoning prompting outperforms traditional fine-tuning and direct prompting methods, achieving human-level translation quality.

Theoretical Contributions: The study contributes to the theoretical understanding of no-resource translation, highlighting the limitations of traditional MT methods and the potential of LLMs in linguistically diverse scenarios.

Practical Implications: The findings have significant practical implications for language preservation, particularly for endangered languages with minimal digital representation.

Cons
Limited Generalizability: The study is based on a single no-resource language (Owens Valley Paiute), and the generalizability of the findings to other languages is not thoroughly explored.

Lack of Synthetic Data Experiments: While the authors mention the potential for synthetic data generation, they do not provide concrete experiments or results in this direction.

Fine-Tuning Failure Analysis: The paper could benefit from a more detailed analysis of why fine-tuning fails in no-resource scenarios, as opposed to low-resource ones.

Prompt Clarity: The specific prompts used for chain-of-reasoning and direct prompting are not provided, which could limit the reproducibility of the study.

Ethical Considerations: The paper does not discuss potential ethical concerns, such as the risk of misrepresentation or cultural insensitivity when translating endangered languages.

Conclusion
Overall, this paper makes a significant contribution to the field of machine translation by addressing the unique challenges of no-resource languages. The innovative use of chain-of-reasoning prompting demonstrates the potential of LLMs to perform high-quality translations with minimal data, offering a promising pathway for language preservation. While the study has some limitations, particularly in terms of generalizability and fine-tuning failure analysis, it provides a strong foundation for future research in this important area.


Suggestions for Improvement
Expand Language Coverage: Future work should include experiments with multiple no-resource languages from different linguistic families to validate the generalizability of the findings.

Synthetic Data Experiments: The authors should explore the use of synthetic data generation to expand the corpus size and improve translation performance.

Detailed Prompt Examples: Providing specific examples of the prompts used for chain-of-reasoning and direct prompting would enhance the clarity and reproducibility of the study.

Ethical Considerations: The paper should address potential ethical concerns related to the translation of endangered languages, including the risk of misrepresentation or cultural insensitivity.

Fine-Tuning Failure Analysis: A more detailed analysis of why fine-tuning fails in no-resource scenarios would provide valuable insights into the limitations of traditional MT methods.

---

### Official Review · Reviewer_MgCR · 2025-03-02
**Towards Neural No-Resource Language Translation: A Comparative Evaluation of Approaches**

**Rating:** 7
**Confidence:** 4

**Review:**

This paper investigates how neural LLMs handle translation for languages with no direct training data. The authors compare different approaches, including fine-tuning, chain-of-reasoning prompting, and direct prompting, to evaluate their effectiveness. The study aims to understand how LLMs generalize translation patterns for no-resource languages and whether intermediate languages play a role in translation performance.

Strengths
Novel Problem Focus: The paper addresses an important and underexplored issue—translation for truly no-resource languages, which is critical for language preservation.
Comprehensive Comparison: The study evaluates multiple translation strategies, providing a well-rounded assessment of different techniques.
Well-Defined Experiments: The methodology is clear, with a structured evaluation framework using LLM-based approaches.
Useful Insights: The findings reveal key differences between fine-tuning, chain-of-reasoning prompting, and direct prompting, offering valuable takeaways for future research.

Weaknesses
Limited Language Diversity: The study focuses on a small set of languages, making it unclear whether the findings generalize to all no-resource languages.
Lack of Dataset Analysis: The paper does not explore whether LLMs leverage indirect exposure to target languages through pretraining data.
No Practical Implementation: While the study identifies effective methods, it does not discuss how these approaches could be deployed in real-world translation systems.

---

### Official Review · Reviewer_kKXF · 2025-03-03
**Interesting application, a bit difficult to read**

**Rating:** 6
**Confidence:** 3

**Review:**

The authors investigate strategies for evoking translation of no-resource languages via decoder-only models. The language of interest is Owens Valley Paiute. The authors explore low-resource translation of this language via fine-tuning and ICL/chain-of-thought with PaLM. The corpus consists of 100 sentences, entering a regime which the authors define as "no-resource." Chain-of-thought shows promising BLEU scores on held-out translations.

The paper is difficult to read mostly because it's not clear whether 100-"words" or 100-"phrases" are used. The authors frequently switch between these terminologies (such as in section 4.1). The provided dataset link is dead. Since some sentences (or words?) are included in the ICL, naturally, this changes the interpretability of the BLEU score as the validation set is changing in size. The authors should have simply held out 20-25 sentences across all settings and then vary the amount of examples given. (If this is already occurring, it's not clear.). Given the limited size of the corpus, it would have been interesting to see how full fine-tuning affected performance vs QLoRA.

I prefer to accept this paper as it fits the theme of the workshop and the machine translation community rarely considers ultra-low/no-resource language at this scale, especially using training-free strategies. If this is accepted, I encourage the author to compare with decipherment, which similarly tries to achieve translation with no/limited parallel corpuses (see Ravi and Knight, "Deciphering Foreign Language." ACL, 2011 -- these two authors have a series of seminal works on this topic.)

---

### Decision · Program_Chairs · 2025-03-04

**Decision:**

Reject

**Comment:**

This paper explores no-resource language translation using LLMs by comparing different approaches. Despite positive reviews, the paper is not aligned with the focus of the Building Trust in LLMs workshop, as it primarily addresses language preservation and translation strategies rather than trust, safety, or alignment in LLMs. Given its low relevance to the workshop, I recommend rejection.